# Identification of 3-Aryl-1-benzotriazole-1-yl-acrylonitrile as a Microtubule-Targeting Agent (MTA) in Solid Tumors

**DOI:** 10.3390/ijms25115704

**Published:** 2024-05-24

**Authors:** Stefano Zoroddu, Luca Sanna, Valentina Bordoni, Lyu Weidong, Sergio Domenico Gadau, Antonio Carta, David J. Kelvin, Luigi Bagella

**Affiliations:** 1Department of Biomedical Sciences, University of Sassari, 07100 Sassari, Italy; szoroddu@uniss.it (S.Z.); lusanna@uniss.it (L.S.); v.bordoni@studenti.uniss.it (V.B.); 2Division of Immunology, International Institute of Infection and Immunity, Shantou University Medical College, Shantou 515031, China; lvweidong1230@126.com (L.W.); dkelvin@jidc.org (D.J.K.); 3Department of Veterinary Medicine, University of Sassari, 07100 Sassari, Italy; sgadau@uniss.it; 4Department of Medicine, Surgery and Pharmacy, University of Sassari, 07100 Sassari, Italy; acarta@uniss.it; 5Department of Microbiology and Immunology, Dalhousie University, Halifax, NS B3H 4R2, Canada; 6Sbarro Institute for Cancer Research and Molecular Medicine, Centre for Biotechnology, College of Science and Technology, Temple University, Philadelphia, PA 19122, USA

**Keywords:** RNAseq, MTA, microtubules, cancer, solid tumors

## Abstract

Recently, a compound derived from recent scientific advances named **34** has emerged as the focus of this research, the aim of which is to explore its potential impact on solid tumor cell lines. Using a combination of bioinformatics and biological assays, this study conducted an in-depth investigation of the effects of **34**. The results of this study have substantial implications for cancer research and treatment. **34** has shown remarkable efficacy in inhibiting the growth of several cancer cell lines, including those representing prostate carcinoma (PC3) and cervical carcinoma (HeLa). The high sensitivity of these cells, indicated by low IC_50_ values, underscores its potential as a promising chemotherapeutic agent. In addition, **34** has revealed the ability to induce cell cycle arrest, particularly in the G2/M phase, a phenomenon with critical implications for tumor initiation and growth. By interfering with DNA replication in cancer cells, **34** has shown the capacity to trigger cell death, offering a new avenue for cancer treatment. In addition, computational analyses have identified key genes affected by **34** treatment, suggesting potential therapeutic targets. These genes are involved in critical biological processes, including cell cycle regulation, DNA replication and microtubule dynamics, all of which are central to cancer development and progression. In conclusion, this study highlights the different mechanisms of **34** that inhibit cancer cell growth and alter the cell cycle. These promising results suggest the potential for more effective and less toxic anticancer therapies. Further in vivo validation and exploration of combination therapies are critical to improve cancer treatment outcomes.

## 1. Introduction

One of the three main parts of the cytoskeleton, microtubules, plays an important role in numerous vital cellular processes and functions [1]. Several cellular processes are reliant on the dynamic nature of microtubules, which are polymers capable of rapid assembly and disassembly [2,3]. Normal dynamic rearrangements of the microtubule network necessary for cellular activities can be prevented by the disruption of microtubule dynamics with the formation of abnormally stable or unstable microtubules [4,5,6,7]. Microtubules are dynamic, constantly contracting and expanding due to the reversible attachment and dissociation of α- and β-tubulin heterodimers [8]. Microtubule breakdown is accompanied by the emergence of a new network of spindle microtubules, which are up to 100 times more active than interphase microtubules. Mitotic spindles are produced as a result of this procedure. In order to ensure chromosome attachment and segregation during cell division, the production of mitotic aster and centromeric microtubules requires strict regulation of microtubule dynamics [9]. Moreover, conventional chromosome segregation is possible due to the intrinsic dynamics of microtubule fibers. Cell cycle arrest at the mitotic checkpoint, triggered by improper chromosome attachment or separation, can lead to apoptosis. This process underscores the critical surveillance mechanisms ensuring accurate chromosome segregation, essential for cellular integrity and function [10]. Microtubule targeting agents (MTAs) are currently among the most efficient drugs for the treatment of solid and hematological cancers [11]. Consequently, tubulin-binding sites capable of interacting with molecules affecting microtubules and tubulin dynamics are distinct and well described [12]. Vinca alkaloids are a class of chemicals that interact with and prevent microtubules from polymerizing [13]. The polymerization and stabilization of microtubules are favored by another class of chemicals, including taxanes [14]. Moreover, a third class of substances, including noscapinoids [15,16], interact with microtubules without significantly promoting or inhibiting microtubule polymerization. However, all of the aforementioned drugs act on the spindle assembly checkpoint to generate mitotic spindle defects and halt mitotic progression [5,7]. This provides the molecular basis for using drugs interfering with microtubules to treat various forms of human cancer with chemotherapy [17,18,19,20].

Our previous research analyzed the mechanism of action of 3-aryl-1-benzotriazole-1-yl-acrylonitrile (initially named ‘compound **1**’ [21] and later referred to as ‘compound **34**’ [22], as in the present study) that showed antiproliferative activity in cancer cells of various tumor types [21,23]. This study helped us discover other derivatives, which were used in further subsequent studies [22,24]. However, the cytotoxic action of the compound 3-aryl-1-benzotriazole-1-yl-acrylonitrile remains markedly higher than that of all derivatives (Figure 1). Therefore, in the current study, its role in interfering with microtubulin function was explored through bioinformatic and biological analyses on different types of solid tumors.

## 2. Results

### 2.1. Cytotoxic Effect of **34** in Different Solid Tumor Cell Lines

**34** was employed in subsequent investigations on solid tumors after explored its role in hematological malignancies [21]. The varied concentrations used to treat HeLa, PC3, MCF7, SKMEL28, SKMES-1 and HepG2 ranged from 62.5 to 1000 nM. After 48 h, a colorimetric experiment was conducted, and Table 1 displays the IC_50_ calculations. In contrast to all the other cell lines, **34** had noticeably greater action on the HeLa and PC3 cells.

### 2.2. **34** Inhibits Colony Formation in Cancer Cells

By using a colony formation assay, the ability of **34** to cause cell death was evaluated. This assay is helpful for assessing the capacity of tumor cell lines to form colonies for determining its potential efficacy in vivo [25]. The selection of the starting concentrations for compound **34** was meticulously planned to capture a broad spectrum of its biological activity. For the HeLa cells, with an IC_50_ value of 20 nM, we commenced at the IC_50_ concentration of 20 nM. This approach aimed to directly assess the compound’s effect at its half-maximal inhibitory concentration. Conversely, for the PC3 cells, which exhibited an IC_50_ of 80 nM, the starting concentration was set at 50 nM. This decision was informed by our objective to explore the compound’s efficacy across a range that not only centers on but also spans slightly above and below the IC_50_ values. Such a strategy was designed to provide a comprehensive understanding of the potency and efficacy of compound **34** across different cellular contexts. In the Hela and PC3 cell lines, the effects were assessed. According to Figure 2, **34** was already able to suppress cell growth at doses below the IC_50_ value after two weeks in PC3 cell line. The same trend is shown for the Hela cell line (Figure 2). The original images are provided as Appendix A showcases the colonies for the HeLa cells, and Appendix A showcases those for the PC3 cells.

### 2.3. **34** Induces Cell Cycle Arrest in G2/M Phase

The distribution of the cell cycle was examined by flow cytometry in order to investigate the molecular effects of the substance on the cancer cell lines HeLa and PC3. By FACS analysis, a significant increase in the G2/M phase was observed compared to the untreated control after treatment with **34** in both cell lines (Figure 3). **34** showed a time-dependent effect with an early phase of action at 6 h and a late phase showing an accumulation of cells in G2/M, already, at 12 h and at 18 h for HeLa. Particularly, the HeLa cells treated for 6 h showed a reduction in the G1 phase and an increase in the G2/M phase of the cell cycle compared to the controls. At 8 h, there was already a clear block of the cell cycle in G2/M that was even more pronounced after 12 and 18 h of treatment. The effect of the **34** was similar in PC3, with an early increase in the S-phase between 6 and 8 h, followed by a late phase at 18 h in which most cells were in a clear block G2/M. The reported data show that the results obtained from the FACS analysis specify the promising antitumor activity of **34**, based on G2/M cell cycle arrest.

### 2.4. Analysis of Differently Expressed Genes after 34 Treatment by RNAseq

Gene expression profiling was performed by RNAseq. Based on the cell cycle results, the HeLa cells were treated with **34** for 6 and 12 h, while the PC3 cells were treated for 8 and 18 h, and sequencing was performed by Novogene Co., Ltd., Beijing, China. After sequencing was complete, the quality of the original data was evaluated, and unqualified reads were eliminated using Trimmomatic to increase the precision and effectiveness of sub-sequencing analysis. The clean readings were acquired, and then, STAR alignment and expression analysis were carried out. DEseq2 was used to analyze differential expression in both the control and experimental group samples based on the sequence alignment. The values of the parameters log2 fold-change and p-value were used as reference indicators for significant differences, setting |log_2_FC| > 2 and the *p*-value < 0.05 as statistically significant DEGs. Based on the screening criteria, after treatment with **34** in the HeLa cells at a concentration of 1 μM compared to the control, we found the following differential genes: 2279 genes were upregulated and 314 were downregulated after 6 h of treatment; 2302 genes were upregulated and 347 were downregulated after 12 h of treatment; and 33 genes were upregulated and 47 genes were downregulated compared 6 h to 12 h of treatment (Figure 4A). In the PC3 cells at a concentration of 1 μM compared to the control, we found the following differential genes: 68 genes were upregulated and 58 genes were downregulated after 8 h of treatment; 125 genes were upregulated and 114 were downregulated after 18 h of treatment; and 98 genes were upregulated and 67 genes were downregulated compared 8 h to 18 h (Figure 4B).

### 2.5. GO Mapping and KEGG Pathway Analyses of DEGs

Online tools were used to conduct GO and KEGG pathway studies to understand the roles of the various genes (OmicShare tools). The various KEGG functions in **34**-treated cells were annotated using GO. The GO enrichment analysis indicated that the biological processes regulated by **34** in HeLa were mainly related to cellular metabolic processes, cellular component organization and protein biogenesis (Figure 5A,B). According to the results of the KEGG pathway analysis, following **34** treatment, DEGs were mainly enhanced in cancers, infectious diseases, endocrine system, cell growth and death, signal transduction, translation, and cell metabolism (Figure 6A,B). Regarding PC3 cells in terms of biological functions, top GO showed similar results to HeLa with enhancement especially in stress response, regulation of cell proliferation, regulation of cell motility, regulation of molecular functions, and components of motility (Figure 7A,B). Based on the results of KEGG pathway analysis, following **34** treatment, GO was improved mainly in cancers, immune system, cell growth and death, signal transduction, replication and repair, and energy metabolism (Figure 8A,B).

### 2.6. Protein–Protein Interaction (PPI) Network Analysis and Hub Genes Screening

We thoroughly examined the PPI network to study the regulatory interaction between DEGs. Distinct protein–protein interaction networks were constructed using the online mapping application Network Analyst at 6 and 12 h for Hela and at 8 and 18 h for PC3 (Figure 9 and Figure 10). The first 15 genes were identified as hub genes per grade by the PPI network analysis. The genes discovered after the last hub for HeLa were TUBA1C, TUBB2A, TUBB4B, TUBA1B, UBC, CDKN1A, ZNF217, MYC, HIST1H2BD, NOTCH3, CCNE2, MCM5, TOP2A, CENPE and CDK1 (Figure 11). The results of the functional and pathway enrichment analysis showed that these 15 hub genes were mainly involved in protein-containing complex subunit organization, chromosome organization, DNA conformation change, nucleosome assembly, metabolic processes, the regulation of gene silencing and the negative regulation of gene expression (Table 2). Similarly, the top nine hub genes for PC3 were identified from the PPI network analysis. The genes discovered were MYC, CDK1, TUBB4B, TUBB2A, TUBA1C, UBC, MCM5, MCM6 and CCNE2 (Figure 12). The results of the functional enrichment and pathway analysis showed that these nine hub genes were mainly involved in mitotic cell cycle phase transition, DNA replication initiation and microtubule-based processes (Table 3). The main signaling pathways involved were the cell cycle, the p53 signaling pathway, cell senescence and DNA replication (Table 3).

### 2.7. Connectivity Map and Drugbank Analysis of DEGs

Positive and negative regulatory gene groups were identified in the differential gene areas obtained after **34** administration on the HeLa and PC3 cells. The gene expression profiles of well-known compounds were compared with those of our molecule using the CMap database (https://www.Broadinstitute.Org/connectivity-map-cmap (accessed on 10 December 2023)), and the degree of correlation was utilized to identify the drug molecules with the highest correlation. Furthermore, the analysis summarized the possible targets and mechanisms of **34** based on similar drug profiles. The results of the CMap analysis showed that **34**-induced DEGs on HeLa were correlated with several inhibitors including tubulin inhibitors. Small molecule compounds in the top 10 connectivity score are summarized in Table 4. Similarly, the results of the CMap analysis on **34**-induced DEGs in the PC3 cells were correlated with different tubulin inhibitors (Table 5).

### 2.8. Compound **34** Induces Tubulin Polymerization

To further confirm that the **34** drug interacts with the tubulin assembly, the tubulin polymerization assay was performed. Figure 13 shows how paclitaxel and **34** affect the process of tubulin polymerization. In particular, the action of compound **34** shows a higher speed in the nucleation process than the already known paclitaxel [43]. These results confirmed that compound **34** significantly increases tubulin polymerization compared with the control. Furthermore, compared with paclitaxel at equimolar concentrations, **34** was able to bind with higher affinity, increasing the rate of polymerization resulting in a shorter stabilization time. Therefore, the **34** compound could interact with the same binding site as paclitaxel and be considered as an MTA.

## 3. Discussion

Despite the enormous progress made in recent years in the field of oncology, the exploration for new potent chemotherapeutic agents remains an ongoing challenge. Indeed, cancer is still among the leading causes of death worldwide. The main tools remain the use of surgery, radiotherapy, and chemotherapy. These techniques are used individually or in combination, but the last of these approaches, although more effective, causes medium to severe systemic side effects. For this reason, research is increasingly focusing on the discovery of new, potent, and selective chemotherapeutic agents that can reduce toxic effects and improve therapy and the quality of life of patients. In recent years, new potential chemotherapeutic agents of natural, synthetic or a combination of the two have been developed [24,44]. Among these, antimitotic drugs play a significant role. Tubulin is a protein component of all cells that is crucial for maintaining cytoskeletal structure, movement, and cell cycle progression. In this process, α- and β-tubulin dimers undergo nucleation and polymerization, joining with other dimers to form the microtubule network and the mitotic spindle, essential for cell cycle completion. As described by Hanahan and Weinberg [45], cancer cells have the potential for unlimited replicative capacity with a deregulated cell cycle resulting in much faster cell progression. This results in a more dynamic microtubular network with frequent polymerization and depolymerization cycles. Therefore, research in recent years has increasingly focused on finding new chemotherapeutic drugs directed against tubulin [46]. MTAs belong to this class of drugs and are mainly divided into two groups: microtubule-destabilizing agents (MDAs) and microtubule-stabilizing agents (MSAs). Paclitaxel is a chemotherapeutic agent belonging to the taxane class and is among the most important MTAs drugs on the market and used in the treatment of advanced ovarian cancer, breast cancer, non-small cell lung cancer, prostate cancer, and AIDS-associated Kaposi’s sarcoma. However, the occurrence of resistance limits treatment options. These drugs could be resistant to them through several pharmacodynamically distinct methods. 

Our work is based on the results obtained by Carta et al. 2011, in which some classes of 3-aryl-1-benzotriazole-1-yl-acrylonitrile derivatives were active against tumor proliferation. Specifically, a combination of experimental and computational results has hypothesized that this class of compounds may act as antiproliferative agents by interacting with tubulin. Through bioinformatic and biological assays we further investigate the action of one of these compounds, (*E*)-2-(1*H*-benzo[*d*][1,2,3]triazol-1-yl)-3-(4-methoxyphenyl)acrylonitrile **34** on some solid tumor cell lines. To clarify what concentrations this drug was able to arrest tumor growth at, an MTT assay was performed on prostate carcinoma, breast carcinoma and cervical carcinoma lines. The concentration of **34** drug capable of reducing the absorbance of treated cells by 50% (IC_50_) compared with untreated cells was calculated on each cell line. Notably, HeLa cells appeared more sensitive than other cell lines with an IC_50_ of 23.78 nM after 48 h of treatment. However, PC3 and MCF7 were also shown to be highly sensitive to treatment at fairly low concentrations with an IC_50_ of 83.02 and 100.8 nM, respectively. Only the cells that showed the most sensitivity to the treatment were included in subsequent cell cycle analyses, specifically HeLa and PC3 cells.

The dysregulation of the cell cycle is a major cause of the tumor initiation phase. Our results showed that **34** can significantly alter the cell cycle distribution ratio of both cell lines and significantly block cell growth in the G2/M phase. Cell cycle arrest can lead to cell death, and when the DNA of cancer cells is damaged, selective blockade in the G2 phase occurs. In this study, flow cytometry was used to detect **34**-treated cell cycle changes. The results suggest that **34** can affect cell proliferation by inducing G2/M phase arrest. 

Computationally, we performed several analyses to evaluate the major genes involved following **34** treatment. Comparing the DEGs obtained from **34** treatment administered to the two cell lines at the two times for each with the gene hubs identified by PPI analysis, it is evident that the results of KEGG and GO enrichment were similar. In our study, compound **34** emerges as a pivotal agent influencing key cellular processes, notably the transition of the biological cycle to mitosis, microtubule-based processes, and the initiation of DNA replication. This initial insight lays the groundwork for a deeper investigation into the compound’s mechanism of action and its potential therapeutic implications. However, the complexity of these cellular effects demands a more nuanced analysis, particularly when considering the DEGs observed in cells treated with compound **34**. A comparative analysis of DEGs in cells treated with compound **34** against both untreated cancer cells and normal cells from corresponding tissues could significantly advance our understanding of the compound’s biological impact. Such an approach, as suggested by Sarhadi et al. (2022) [47], offers a pathway to discern whether compound **34**’s influence on gene expression could drive cancer cells toward a gene expression profile resembling that of non-cancerous cells, an outcome that would highlight its potential as a therapeutic agent. While the present study has not encompassed this broad comparative framework, our future research is poised to delve into this arena. By examining how compound **34** alters the gene expression profiles of cancer cells in relation to their non-cancerous counterparts, we aim to unveil the extent to which the compound may normalize aberrant cancer cell phenotypes. Moreover, compound **34**’s capacity to modulate various critical pathways, including the cell cycle, DNA replication and the p53 signaling pathway, suggests its broad-spectrum influence on cellular pathways that can culminate in cancer cell death. The protein–protein interaction (PPI) network analysis further elucidates compound **34**’s effect, revealing notable similarities in protein expression alterations between HeLa and PC3 cell lines, echoing findings by Hozhabri et al. (2022) [48] on the interconnectedness of these pathways in cancer pathogenesis. By bridging these molecular insights with comparative gene expression analysis, our future studies will seek to illuminate the full spectrum of compound **34**’s impact on cancer cells, potentially offering a paradigm shift in how we approach the normalization of cancer cell behavior through therapeutic interventions. In fact, both cell lines after treatment were affected in the expressions of TUBB4B, TUBB2A, TUBA1C, MCM5, UBC, CDK1, CCNE2 and MYC. All these genes are involved in crucial biological processes. TUBB4B, TUBB2A and TUBA1C are cellular components of the large family of microtubule proteins. Alterations at various levels of some of these proteins can cause pathological conditions. Particularly, some of these processes are caused by the malfunction of polymerization and depolymerization of microtubules, inducing a condition called “dynamic instability” [49]. Studies have shown that TUBB4B plays a role in the maintenance of cancer stem cells (CSCs) in oral cancer. The protein is expressed in most normal tissues by cytoplasmic expression and is detected in all cell lines and cancer tissues [50]. In some cancers such as glioma, carcinoid, and cholangiocarcinoma, weak to moderate staining of the protein is observed [50]. Tubulin heterogeneity, which includes TUBB4B, has also been linked to the development of drug resistance in cancer [51]. TUBB2A has been identified as a potential target for cancer therapy due to its role in cancer progression. Several studies have reported an association between TUBB2A expression and poor prognosis in various cancer types, including breast cancer, pancreatic cancer, and non-small cell lung cancer [51]. The precise mechanism by which TUBB2A promotes cancer progression is still under investigation. One study found that TUBB2A overexpression led to increased microtubule stability and resistance to paclitaxel-induced cell death in breast cancer cells [52]. Another study showed that TUBB2A overexpression promoted the formation of filopodia, which are involved in cancer cell migration and invasion [53]. Targeting TUBB2A may represent a promising approach for cancer therapy. Several drugs that target microtubules, such as paclitaxel and vincristine, have been approved for cancer treatment. However, these drugs have limited efficacy and can cause severe side effects. TUBA1C plays a role as a tumor promoter in most cancer types [54]. The overexpression of TUBA1C has been correlated with poor prognosis in patients with lung adenocarcinoma (LUAD) and is associated with 13 types of tumor-infiltrating immune cells (TIICs) in the tumor microenvironment [55]. Additionally, TUBA1C has been shown to play an important role in cancer immunity, as its expression positively correlates with stromal and immune cells [56]. In glioma, TUBA1C has been linked to poor prognosis and its potential oncogenic mechanisms are currently being investigated [56]. Overall, TUBA1C appears to be involved in the development and progression of various types of cancer and may serve as a potential prognostic biomarker and therapeutic target. The MCM5 gene, while not directly involved in the assembly of the microtubule network, is important for the maintenance of the cellular chromosome compartment. It is overexpressed in various types of cancer, including breast, colorectal and pancreatic cancers, and its expression levels have been correlated with poor prognoses in these cancers [57]. UBC is a gene that encodes for a polyubiquitin precursor protein, which plays a critical role in protein degradation through the ubiquitin-proteasome system (UPS). The UPS has been linked to several illnesses, including cancer. It has been demonstrated that UBC conjugation to other molecules has an impact on biological processes like cell cycle control, DNA repair, and other cell signaling networks [58]. UBC expression is widespread across a variety of organs. Breast, ovarian, and colorectal cancers are just a few of the cancer kinds in which UBC has been discovered to be overexpressed [59]. Interestingly, rather than transcription, a negative feedback process linking UBC gene translation to ubiquitin levels was discovered. This process could be involved in controlling the expression of UBC in cancer cells. Overall, it appears that UBC, by controlling protein degradation through UPS, plays a significant role in cancer initiation and evolution. CDK1 is a member of the cyclin-dependent kinase family of proteins and is primarily involved in the regulation of the cell cycle during both mitosis and meiosis. CDK1 has been shown to interact with several tumor suppressor proteins, such as p53 and Rb, and its overexpression has been linked to the development of various cancers, including breast, lung, and colorectal cancers [60]. In preclinical research, the use of CDK1 inhibitors has been explored as a possible therapeutic approach for cancer therapy. Recent research has also highlighted the importance of CDK1 in other cellular processes, such as transcription, epigenetic control and stem cell self-renewal, which may contribute to its role in cancer initiation and spread. Overall, dysregulation of CDK1 is an important target for cancer treatment and is essential for its growth.

The protein CCNE2, also referred to as Cyclin E2, is essential for controlling the cell cycle, especially during the G1/S phase shift. It attaches to CDK2 and stimulates it similarly to Cyclin E1, creating an active kinase complex [61]. Leukemia and stomach cancer are two cancers with which dysregulated Cyclin E2 expression has been linked. Finally, MYC plays a role in the regulation of division and cell growth, and its overexpression has been linked to the development and progression of cancer. MYC antagonizes p27, a key cell cycle regulator, and this antagonism plays a crucial role in human cancer development [62].

This study of the effects of **34** involved the use of CMap to identify high correlation drug molecules, which were then analyzed to summarize their possible targets and mechanisms of action. Correlation is a statistical method used to evaluate the association between two variables, and it provides a measure of the strength and direction of the relationship. In statistics, there are different types of correlation measures, such as the Pearson correlation, Kendall rank correlation, Spearman correlation and Point-Biserial correlation, each with its assumptions and applications. Therefore, studying the effects of **34** using CMap highlights the importance of correlation analysis in identifying potential drug targets and mechanisms of action. The results showed that **34**-induced DEGs were mainly related to tubulin inhibitors, growth inhibitors and PLK inhibitors. Based on the identified hub genes and small molecule targets obtained by CMap comparison, possible **34** targets could be TUBB and TUBA4A. Therefore, the results strongly suggest that the target of **34** may be tubulin and the cellular microtubule network. The normal structure and function of the microtubules may be affected when drugs attach to certain tubulin sites, which could affect the assembly. The internal balance of cells is disrupted when microtubules are damaged, resulting in cell cycle arrest and apoptosis [63]. In the current study, we explored how **34** can influence tubulin polymerization, resulting in a cell blockage in the G2/M phase of the cell cycle. This phenomenon is akin to the action of other drugs known to affect tubulin assembly, such as paclitaxel. Indeed, it was noted that **34** has the ability to bind similarly to paclitaxel to microtubules. Specifically, the analysis revealed that the rate of tubulin nucleation between the α and β subunits appeared to be faster when treated with **34** compared to equimolar concentrations of paclitaxel and the negative control. This implies that **34** accelerates the process of microtubule nucleation, a crucial step in microtubule formation during the M phase of the cell cycle. These findings further confirm the potential antimitotic effect of our compound. The acceleration of microtubule nucleation and the resulting blockage of the G2/M cell cycle phase indicate that **34** may represent a new and potent chemotherapeutic weapon in the fight against cancer. The paclitaxel-like effect and the higher binding affinity of **34** are promising indicators of its potential as an antitumor agent, although further research and clinical studies are required to fully confirm its efficacy and safety in cancer treatment.

In conclusion, our study has demonstrated that **34** possesses a multifaceted mechanism of action, including the ability to induce cell cycle arrest in the G2/M phase, disrupt tubulin polymerization, inhibit cell division, and activate the p53 signaling pathway, ultimately promoting cell death. These results, in conjunction with previous research [21], suggest that **34** holds significant promise as a microtubule-targeting agent in the treatment of cancer. However, further studies are needed to investigate the molecular and systemic pathways of **34** in vivo, including its pharmacokinetics, toxicity, and efficacy in animal models and human clinical trials. Additionally, exploring the potential combination of **34** with other therapies may provide a synergistic effect and improve treatment outcomes for cancer patients.

## 4. Materials and Methods

### 4.1. Cell Culture

HeLa (cervical carcinoma), PC3 (prostate adenocarcinoma) and MCF7 (breast adenocarcinoma) cell lines (ATCC, Rockville, MD, USA) were cultured in Dulbecco’s Modified Eagle’s Medium (DMEM) (Gibco, New York, NY, USA). DMEM was supplemented with 10% of fetal bovine serum (FBS, Gibco, New York, NY, USA), 100 µg/mL streptomycin (Gibco, New York, NY, USA), 1% of L-glutamine and 100 units/mL of penicillin. Then, the cells were incubated at 37 °C with 5% of CO_2_ in a humified incubator.

### 4.2. MTT Assay

The 3-(4,5-dimethylthiazol-2-yl)-2,5-diphenyltetrazolium bromide (MTT) (Sigma-Aldrich, St. Louis, MO, USA) cellular reduction level has been quantified. The test compound was added after 24 h of preincubation to achieve a final concentration in the range of 62.5–1000 nM, and the cells were then incubated for 48 h. The MTT was added at the end of this time to a final concentration of 0.5 mg/mL, and the cells were then incubated for 4 h at 37 °C. After removal of the culture medium, the MTT-produced formazan crystals were dissolved by adding 200 µL of DMSO to each well, and the absorbance was measured at 570 nm using a microtiter spectrophotometer. All the experiments were performed in triplicate.

### 4.3. Cell Cycle Analysis

The cells were placed on six wells and, after 24 h, were treated with 1 μM **34** for times of 6-8-12-18 h for HeLa and PC3. The cells were centrifuged at 3000 rpm for 5 min, washed in PBS, fixed in cold 70% ethanol, and incubated at −20 °C overnight. Prior to analysis, the fixed cells were washed twice with PBS, resuspended and incubated with RNAse 100 µg/mL for 30 min at 37 °C and finally incubated in the dark at room temperature with 50 µg/mL PI for at least 20 min. A flow cytometric analysis was performed using a flow cytometer (BD FACSCanto II) that collected 10,000 events, and the data were analyzed using ModFit Lt 6.0 software. All the experiments were performed in triplicate.

### 4.4. Colony Formation Assay

A total of 100–200 cells/well were seeded in a six-well plate and incubated at 37 °C for 2 to 3 weeks, until the cells in the control plates formed visible colonies. Subsequently, the cells were treated, and the culture medium with the compound was changed every three days. The cells were stained with 1 mL of a mixture of 6.0% glutaraldehyde and 0.5% crystal violet for 30 min. The stained colonies were compared to control cells and analyzed with ImageJ. All the experiments were performed in triplicate. In conducting our cell cycle analysis, we employed a concentration of compound **34** significantly higher than its IC_50_ value for HeLa cells. This methodological choice was aimed at capturing the immediate effects of compound **34** on cell cycle dynamics. Employing a higher concentration allowed us to detect subtle changes and ensure a comprehensive exploration of the compound’s impact on cell cycle arrest.

### 4.5. RNAseq Library Preparation and Sequencing

Six-well plates were plated with HeLa and PC3 cells. After 6 and 12 h for the HeLa cells and after 8 and 18 h for the PC3 cells, 1 µM **34** and fresh new medium were added. After carefully removing the supernatant, the cells were washed twice with PBS. The pellet was obtained after treatment with **34** and rinsed twice with PBS after being centrifuged at 1000 rpm for 5 min. The supernatant was then carefully removed. Using the RNeasy MINI kit, total RNA (Qiagen, Hilden, Germany) was extracted. The Agilent 2100 bioanalyzer (Agilent Technologies, Carpinteria, CA, USA) was used to assess the quality of total RNA. Digestion with RNaseH was used to separate ribosomal RNA from the total RNA, and VAHTSTM RNA Clean Beads were used to purify ribosome-deprived RNA. Using random primers, complementary DNA (cDNA) was produced by reverse transcription. DUTP was added for tagging when the second strand of cDNA was produced. After purification of the 150–200 bp fragments with magnetic beads, the libraries were sequenced on the Hiseq-PE150 (Vazyme Biotech Co., Ltd., Nanjing, China). For the RNAseq experiment, the concentration of compound **34** used was notably above the IC_50_ value identified for HeLa cells. This approach was adopted to maximize the transcriptional response to compound **34**, facilitating the identification of a broad range of genes and pathways potentially regulated by the compound. The higher concentration was crucial for revealing the extensive spectrum of biological activities of compound **34**, beyond its cytotoxic capacity.

### 4.6. Differentially Expressed Gene Analysis (DEGs)

The sequencer transformed each data point produced into raw reads. The clean reads were obtained by removing low-quality reads or reads with unknown adapters or bases from the raw data to ensure the accuracy and reliability of the data analysis. The clean reads were matched to reference sequences before using RSEM to calculate gene and transcript expression levels. EdgeR was used to perform the differential intergroup expression analysis.

### 4.7. GO Enrichment and Kyoto Encyclopedia of Genes and Genomes (KEGG) Pathway Analysis for DEG Characterization

Pathway enrichment and functional studies of DEGs were carried out to clarify the biological procedure and molecular mechanism of DEGs. Functional and pathway enrichment was carried out using KEGG pathway analysis and Gene Ontology database (GO) enrichment (omicshare).

### 4.8. Hub Genes and Key Modules

A gene network analysis was performed using STRING (https://string-db.org/cgi/input.pl (accessed on 10 December 2023)), a database of experimentally predicted and confirmed protein–protein interactions. STRING was used to find important hub genes and modules. By combining the results of a differential expression analysis with database interactions and DEGs to create an interaction network, the system employs a scoring mechanism to evaluate the results produced by the various approaches.

### 4.9. Connectivity Map (CMap) and Drugbank Analysis of DEGs

CMap (https://clue.io/cmap (accessed on 10 December 2023)) has been widely used to explore drug repurposing, drug lead discovery, modes of drug action and other topics. After being separated into groups of positive and negative regulatory genes, the gene expression profiles obtained from HeLa and PC3 cells treated with **34** in vitro were compared with the gene expression profiles of substances in the CMap database. Highly correlated drug molecules were identified according to the degree of correlation of the drug molecules and potential targets, and the mechanisms of the drug molecules were listed.

### 4.10. Tubulin Assay

The tubulin polymerization assay was performed using the kit supplied by Cytoskeleton (BK006P, Cytoskeleton, Denver, CO, USA). It is based on the principle that light is scattered by the microtubules in proportion to the polymer concentration of the microtubules. The absorbance was recorded at 340 nm at 37 °C for 60 min with fixed acquisitions every 30 s. The assay was performed by incubating equimolar concentrations of tubulin with paclitaxel and **34**. After 60 min, the optical density of tubulin polymerization was recorded by reading the absorbance.

### 4.11. Datasets

The data are presented as mean ± SD of the replicates. All the experiments were performed at least in triplicate. The data were analyzed using Student’s *t*-test and differences were considered significant at *p* < 0.05. The data analysis for flow cytometry was performed with Modfit Lt 6.0 software.

## Figures and Tables

**Figure 1 ijms-25-05704-f001:**
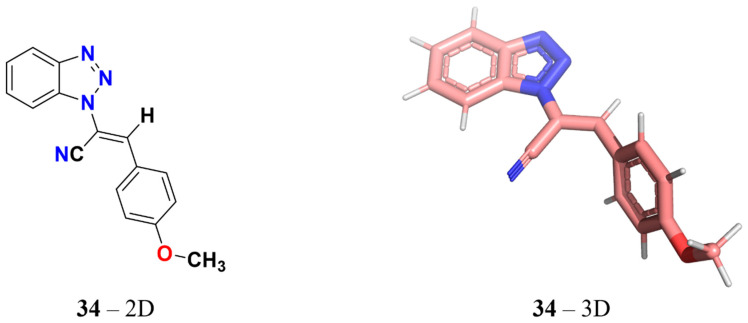
(*E*)-2-(1*H*-benzo[*d*][1,2,3]triazol-1-yl)-3-(4-methoxyphenyl)acrylonitrile (**34**).

**Figure 2 ijms-25-05704-f002:**
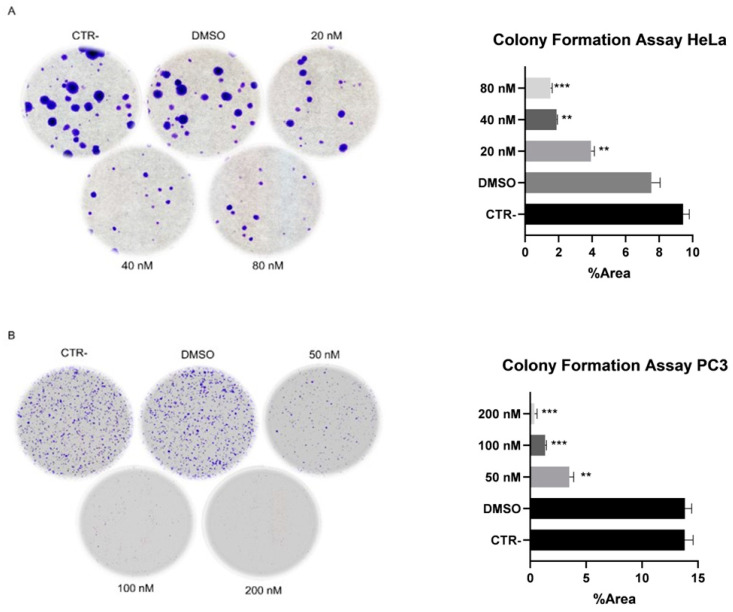
Representative photographs demonstrating effects of **34** exposure on colony formation. HeLa (**A**) and PC3 (**B**) cells were treated with DMSO and **34** at concentrations slightly below and above their respective IC_50_. Colonies were fixed and stained with glutaraldehyde and crystal violet two weeks after cell treatment. Data were analyzed using Student’s *t*-test, ** *p* value < 0.01, *** *p* value < 0.001.

**Figure 3 ijms-25-05704-f003:**
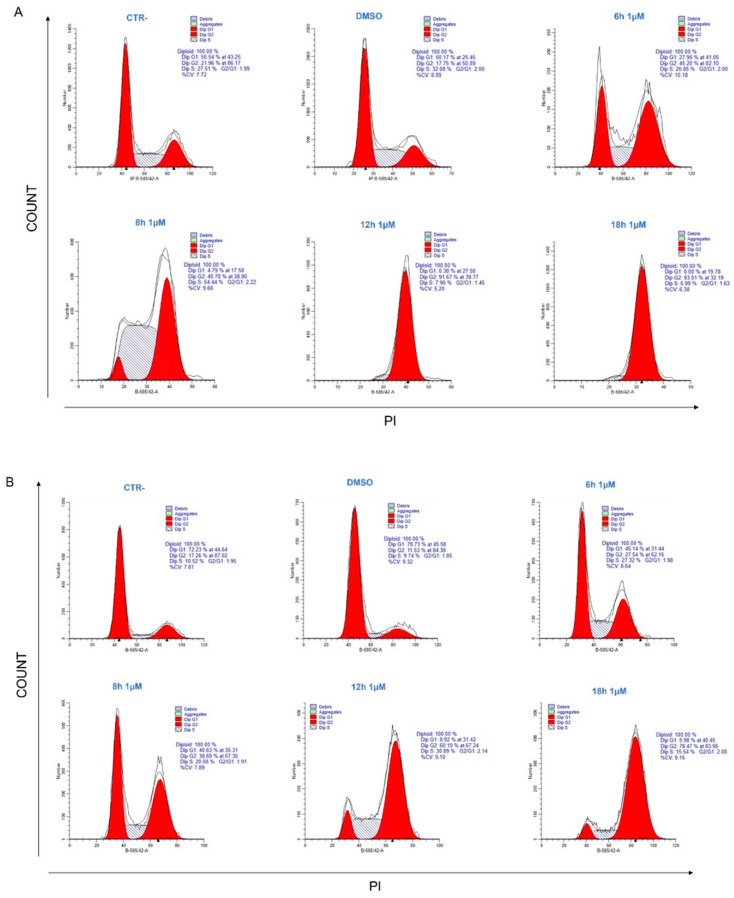
The cell cycle distribution of HeLa (**A**) and PC3 (**B**) cell lines after treatment with **34**. The profiles were obtained by flow cytometry using the PI intercalating dye, with the aim of monitoring cell cycle progression based on DNA quantification.

**Figure 4 ijms-25-05704-f004:**
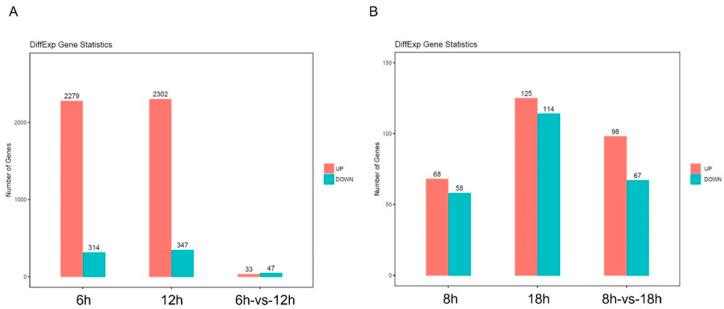
Histogram of differentially expressed genes. Set of differentially expressed genes disturbed by **34** (|logFC| ≥ 2 and *p* < 0.05) in Hela (**A**) and in PC3 (**B**) cells. Genes with significant differential expression are indicated by red (upregulated) and blue (downregulated) picks; (|logFC| ≥ 2 and *p* < 0.05).

**Figure 5 ijms-25-05704-f005:**
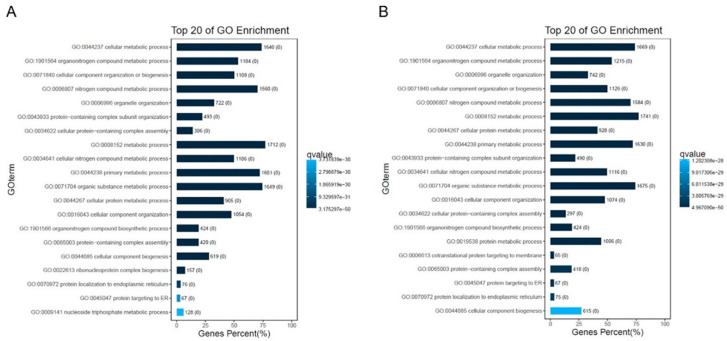
Top 20 GO enrichment analysis of DEGs for different hours in HeLa cells. (**A**) Number of enriched genes after 6 h of treatment. (**B**) Number of enriched genes after 12 h of treatment.

**Figure 6 ijms-25-05704-f006:**
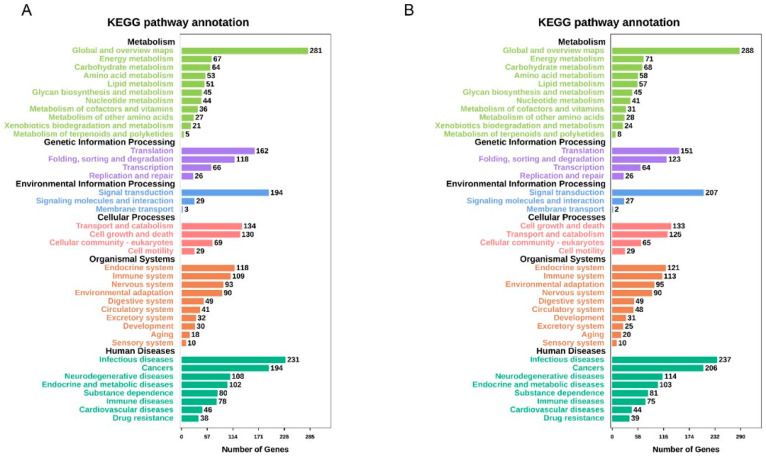
KEGG pathway enrichment analysis of DEGs in HeLa cells. (**A**) KEGG pathway analysis of 6 h treatment DEGs. (**B**) KEGG pathway analysis of 12 h treatment DEGs. Number of enriched genes and pathway terms is presented.

**Figure 7 ijms-25-05704-f007:**
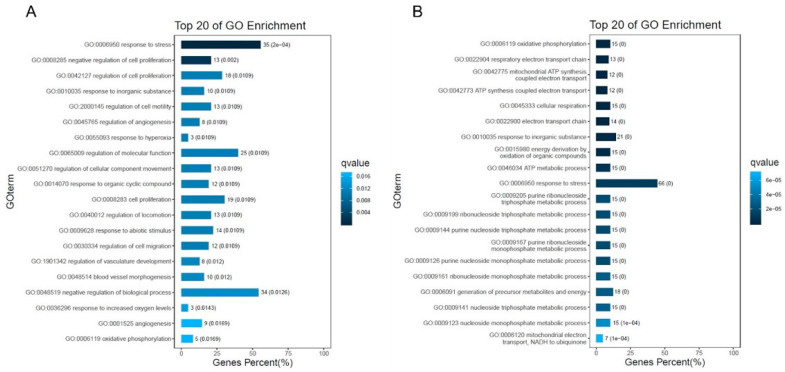
Top 20 GO enrichment analysis of DEGs for different hours in PC3 cells. (**A**) Number of enriched genes after 8 h of treatment. (**B**) Number of enriched genes after 18 h of treatment.

**Figure 8 ijms-25-05704-f008:**
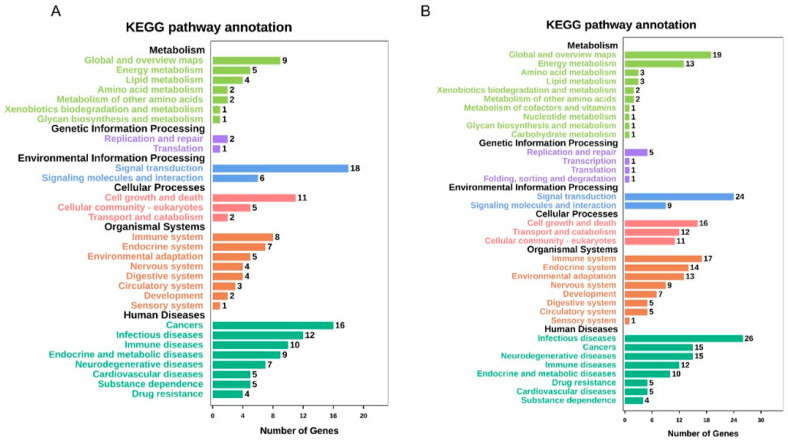
KEGG pathway enrichment analysis of DEGs in PC3 cells. (**A**) KEGG pathway analysis of 8-hour treatment DEGs. (**B**) KEGG pathway analysis of 18-hour treatment DEGs. Number of enriched genes and pathway terms is presented.

**Figure 9 ijms-25-05704-f009:**
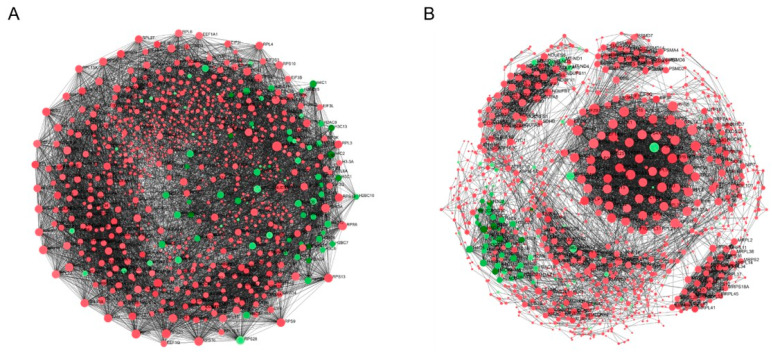
PPI interaction network of DEGs. (**A**) DMSO vs. 6 h, (**B**) DMSO vs. 12 h. Nodes represent differential genes, connections between nodes represent gene interactions, node size represents gene degree; red nodes represent upregulation, green nodes represent downregulation.

**Figure 10 ijms-25-05704-f010:**
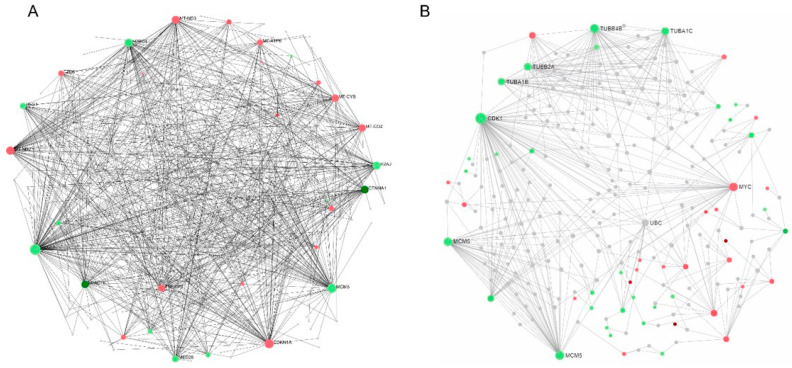
The PPI interaction network of DEGs. (**A**) DMSO vs. 8 h, (**B**) DMSO vs. 18 h. Nodes represent differential genes, connections between nodes represent gene interactions, node size represents gene degree; red nodes represent upregulation, green nodes represent downregulation.

**Figure 11 ijms-25-05704-f011:**
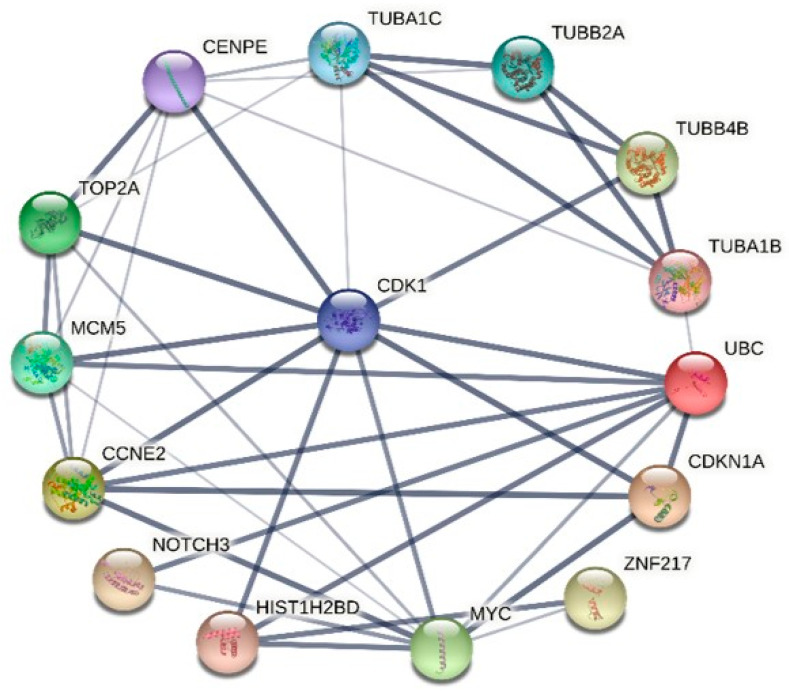
The PPI network of the 15 major hub genes in **34**-treated HeLa cells. The nodes of the network represent proteins; the edges represent PPI.

**Figure 12 ijms-25-05704-f012:**
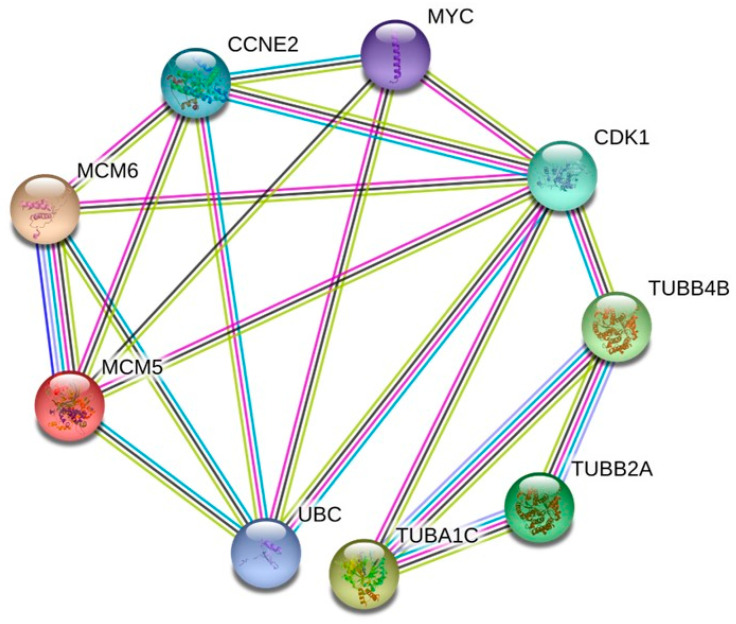
The PPI network of the 9 major hub genes in **34**-treated PC3 cells. The nodes of the network represent proteins; the edges represent PPI.

**Figure 13 ijms-25-05704-f013:**
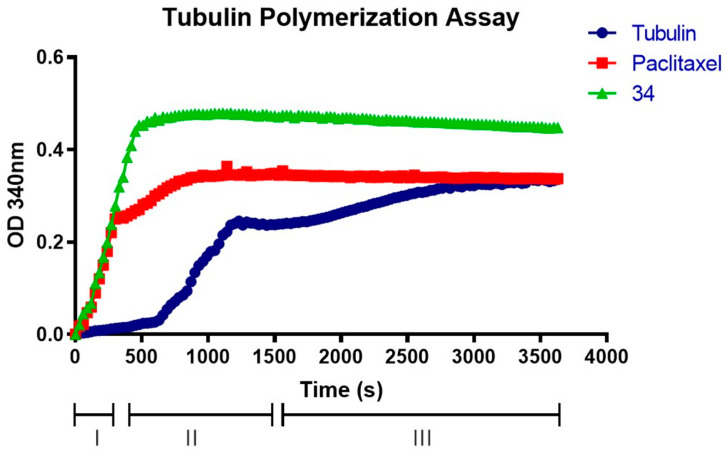
Tubulin polymerization assay. The figure shows a standard polymerization curve (tubulin curve) containing a 100 µL volume of 3 mg/mL tubulin in 80 mM PIPES pH 7.0, 0.5 mM EGTA, 2 mM MgCl_2_, 1 mM GTP and 10% glycerol. The three phases of polymerization of tubulin: I (nucleation), II (growth), and III (steady state). The treated tubulin was pre-incubated with an equimolar concentration of paclitaxel and **34** (10 µM). Then, the polymerization of tubulin was assessed by measuring the absorbance every 30 s for a total of 60 min.

**Table 1 ijms-25-05704-t001:** Determination of cytotoxic activity and IC_50_ values of **34** against different solid tumor cell lines after 48 h.

IC_50_ ^a^ 34 (µM) Mean ± SD	Cell Lines
* 0.2 ± 0.0173	SKMEL28 ^b^
* 0.1 ± 0.0141	MCF7 ^c^
* 0.6 ± 0.0169	SKMES-1 ^d^
* 0.8 ± 0.0412	HepG1 ^e^
0.02 ± 0.0071	HeL ^af^
0.08 ± 0.0264	PC3 ^g^

^a^ Compound concentration (µM) required to reduce cell proliferation by 50%, as determined by MTT method, under conditions allowing untreated controls to undergo at least three consecutive rounds of multiplication. ^b^ Human skin melanoma. ^c^ Human breast adenocarcinoma. ^d^ Human lung squamous carcinoma. ^e^ Human hepatocellular carcinoma. ^f^ Human cervix carcinoma. ^g^ Human prostate carcinoma. * IC_50_ values similar to results obtained in previous work [21].

**Table 2 ijms-25-05704-t002:** Summary of differentially regulated subnetworks disturbed by **34**.

Subnetwork Number	No. of Nodes	GO-BP	False Discovery Rate
1	103	protein-containing complex subunit organization	1.69 × 10^−6^
2	92	chromosome organization	1.03 × 10^−15^
3	40	negative regulation of gene expression, epigenetic	6.49 × 10^−28^
4	49	DNA conformation change	1.56 × 10^−18^
5	24	regulation of gene silencing	2.55 × 10^−10^
6	376	metabolic process	2.61 × 10^−5^
7	47	nucleosome assembly	7.55 × 10^−26^

**Table 3 ijms-25-05704-t003:** Functional and pathway enrichment analyses of top hub genes.

Category	Term	Description	Gene Count	*p* Value
Biological Process (GO)	GO:0044772	mitotic cell cycle phase transition	6	3.34 × 10^−7^
Biological Process (GO)	GO:0000082	G1/S transition of mitotic cell cycle	6	3.29 × 10^−5^
Biological Process (GO)	GO:0006270	DNA replication initiation	3	4.52 × 10^−5^
Biological Process (GO)	GO:0007017	microtubule-based process	5	0.0054
KEGG_PATHWAY	hsa04110	cell cycle	5	6.27 × 10^−8^
KEGG_PATHWAY	hsa03030	DNA replication	2	0.0010
KEGG_PATHWAY	hsa04115	p53 signaling pathway	2	0.0029
KEGG_PATHWAY	hsa04218	cellular senescence	2	0.00040

**Table 4 ijms-25-05704-t004:** Pharmacological disruptors linked to differentially expressed genes induced by **34** in HeLa cells.

Score	Target	Name	Description
99.51	GSK3	Alsterpaullone [26]	Glycogen synthase kinase inhibitor
98.03	p53-HDM-2	RITA [27]	MDM inhibitor
97.67	Cytochrome P-450 51	Econazole [28]	Bacterial cell wall synthesis inhibitor
97.5	β-adrenergic receptors	Cimaterol [29]	Adrenergic receptor agonist
97.37	AChEIs	Huperzine-A [30]	Acetylcholinesterase inhibitor
97.19	HSP70	Evodiamine [31]	ATPase inhibitor
96.51	PDGF	Tyrphostin-AG-1295 [32]	PDGFR receptor inhibitor
95.95	MEK1, MEK2	U-0124 [33]	MEK inhibitor
95.77	β-tubulin	ABT-751 [34]	Tubulin inhibitor
95.74	*p38 MAPK*	TAK-715 [35]	p38 MAPK inhibitor

**Table 5 ijms-25-05704-t005:** Pharmacological disruptors linked to differentially expressed genes induced by **34** in PC3 cells.

Score	Target	Name	Description
99.68	PDE-3, PDE-4, PDE-5	Amrinone [36], Tadalafil [37]	Phosphodiesterase inhibitor
99.50	PLK1, PLK3	Volasertib [38]	PLK inhibitor
99.48	TUBB, TUBA4A	Vincristine [39]	Tubulin inhibitor, microtubule inhibitor
99.07	HIF1A	Flavokavain-B [40]	Carcinoma cell growth inhibitor, hypoxia inducible factor inhibitor
99.00	TUBB	Flubendazole [41]	Acetylcholinesterase inhibitor, microtubule inhibitor
98.28	ACE, CXCL8, IL6, SIRT1, SRD5A1, SRD5A2, TNF	Butein [42]	Angiotensin-converting enzyme inhibitor, EGFR inhibitor, interleukin synthesis inhibitor, NFkB pathway inhibitor

## Data Availability

Data sharing is not applicable to this article.

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
