# Peer review of "Identification of 3-Aryl-1-benzotriazole-1-yl-acrylonitrile as a Microtubule-Targeting Agent (MTA) in Solid Tumors"

_ijms, 2024, doi:10.3390/ijms25115704_

Round 1
Reviewer 1 Report
Comments and Suggestions for Authors
The manuscript entitled „Identification of 3-aryl-1-benzotriazole-1-yl-acrylonitrile as a Microtubule-Targeted Agent (MTA) in solid tumors” describes a study of the anti-tumor activity of a compound previously described as a potential anti-tumor drug, -aryl-1-benzotriazole-1-yl-acrylonitrile (called also 34) using different tumor cell lines, mostly HeLa and PC3. The development of new compounds and their in-deep analysis have a great value in efforts to fight the cancer, and from this point of view the Authors work is important and the manuscript is potentially of interest to the Readers of International Journal of Molecular Sciences.
However, in my opinion, data presented in this manuscript are not yet ready for publication. Below, please find my specific comments regarding the issues that must be revised by the Authors.
1. The title indicated that the presented data show the effect of studied compounds on the microtubular network in solid tumors. However, the presented studies were performed mainly on two cell lines, HeLa and PC3, with no reference to their microtubular cytoskeleton, but only proliferative and colony formation capacities under the treatment with 34. The other result does not support convincingly the thesis that 34 indeed is a microtubule-targeting agent (MTA).
Note also that the title contains an error: instead of “Microtubule-Targeted Agent” it should be “Microtubule-Targeting Agent”.
2. Authors claim that the aryl-1-benzotriazole-1-yl-acrylonitrile was previously published as compound 34 but (if I am not wrong) it seems to me that in the paper Carta et al., 2011 it was published as compound 1. This issue should be clarified for the convenience of the Readers.
3. In the first table and in the first paragraph Authors present data concerning cytotoxicity of aryl-1-benzotriazole-1-yl-acrylonitrile in different cell lines. However, data for the first 4 cell lines, SKMEL28, MCF7, SKMES-1, and HepG1 were already presented in other Authors’ papers (Carta et al., 2002, Carta et al., 2011), therefore it should be mentioned, which results presented in this manuscript are similar to the already published.
4. The colony formation assay: based on the description of the method it is not clear when (before or after plating) and for how long the cell were treated with 34. Thus, it is difficult to refer to this result and compare with other similar findings. I would also suggest to analyze and describe not only the area covered by colonies but also the colony number, which is greatly reduced in samples treated with 34.
5. On the Fig. 3 Authors present the cell cycle analysis. Please, explain why the Authors used such a high concentration of the 34 in this analysis (IC50 for HeLa cells is 50 times lower). The same question raises in the case of the RNAseq experiment. Explaining this issue would be helpful for the Readers.
6. Analysis of the differentially expressed genes (DEGs) is very interesting, but according to the present publication standards, the list of genes with decreased and increased expression level should be included (usually Authors add the table as Supplementary Data). The analysis presented in Fig. 5 and Fig. 6 is very general and not very informative, if not supported by additional experiments, and could be shifted to the Supplementary Data.
I think the analyses of the results regarding DEGs analysis should be improved. For example, Authors state that between DEGs there are some tubulins that in certain cancers are upregulated. How does it refer to the results obtained in this experiment? What can be found in the literature regarding the level of the DEGs identified by Authors in non-treated HeLa and PC3 cells with respect to the non-cancer cell of the cervical and prostate epithelia? Such control would be very informative and could help to answer many questions concerning data obtained in the RNAseq experiments. For example, the comparison of tubulin level between normal and cancer cells and the 34-treated cancer cells could give a view of whether the level of the selected tubulins after the treatment with 34 tend to be normalized to that of normal cells, etc.
The long description of the proteins encoded by DEGs in the Discussion section seem rather a review of the literature than a comparison of results obtained by Authors and previously published results. As mentioned above, I did not find any information of the RNA level (decrease or increase) in 34-treated cells of the proteins described in the Discussion, while this information (and the discussion of this results) would be very informative for the Readers. This information is somehow visible on Figure 11B, but the Figure 9 and 11 are not easily readable.
7. The analysis of tubulin polymerization is my great concern. In previous studies, compounds with a similar structure were shown to decrease tubulin polymerization and to compete with colchicine, suggesting that this type of molecule binds to the colchicine-binding site and function as microtubule-destabilizing agents (MDA). In this view, the result obtained by Authors is contradictory with previous findings. However, this analysis lacks a crucial control, in which the tubulin is treated with the DMSO at the same concentration as in the experimental samples (34 or paclitaxel-containing). DMSO is a known agent that increases the tubulin polymerization in vitro, therefore the lack of this control makes impossible to interpret results obtained by Authors. The analysis of the microtubular network (immunofluorescence) under the treatment with 34 with respect to DMSO control could help to understand if 34 is an MTA and if it functions as MSA or MDA. The very high concentration of both, 34 and paclitaxel in this study should also be explained.
Author Response
Detailed Response to Reviewers:
- Thank you for your invaluable feedback, which has highlighted the necessity for a clearer delineation of how our findings support the thesis that compound 34 acts as a microtubule-targeting agent (MTA) in the studied solid tumor cell lines. We also acknowledge the suggested correction in the title; thus, it will be revised to "Identification of 3-aryl-1-benzotriazole-1-yl-acrylonitrile as a Microtubule-Targeting Agent (MTA) in Solid Tumors". To clarify the specificity of 34's action on microtubules, we would like to point out that in addition to its capability to inhibit cell proliferation and colony formation in HeLa and PC3 cell lines, we have previously cited two pivotal studies in our manuscript (doi: 10.1016/j.ejmech.2021.113590 and doi: 10.1016/j.ejmech.2011.06.018). These studies provided strong evidence through binding assays showcasing the molecule's capacity to interact with microtubules. We have further corroborated this mechanistic action of 34 through a tubulin polymerization assay, as reported in the "Tubulin Assay" section of our manuscript, which confirms 34's direct interference with microtubule dynamics, akin to the known microtubule-stabilizing agent paclitaxel. Moreover, acknowledging the constructive feedback provided, we are planning additional experiments aimed at directly observing the effect of 34 on the microtubule cytoskeleton via immunofluorescence techniques. These planned studies will enable us to visually assess alterations in the microtubule network in treated cells as compared to controls, thereby providing more concrete evidence of 34's role as an MTA.
- Thank you for pointing out the discrepancy in the nomenclature of aryl-1-benzotriazole-1-yl-acrylonitrile between Carta et al., 2011 ("compound 1") and Riu et al. To clarify this for our readers, we've added a note in our manuscript where these studies are first cited. This note explains our use of "compound 34" following the recent nomenclature by Riu et al., and acknowledges the original designation as "compound 1" by Carta et al., 2011. This addition aims to enhance clarity and reader comprehension regarding the naming convention used in our study.
- In response to your comment on cytotoxicity data, we've added a note at line 184 in the table of IC50 values, clarifying that results for SKMEL28, MCF7, SKMES-1, and HepG1 were previously reported in Carta et al. 2011.
- Thank you for your comments regarding the colony formation assay. To clarify, as outlined in our manuscript's Materials and Methods section, the protocol involves allowing cells to grow for two to three weeks before initiating treatment with compound 34. Following this period, the treatment with compound 34 is applied, and the medium with compound 34 is refreshed every three days. This detailed methodology ensures a comprehensive evaluation of compound 34's effects on colony formation. We trust this correction addresses your concerns and further enhances the clarity and accuracy of our manuscript.
- We appreciate your inquiry regarding the use of a higher concentration of compound 34 in the cell cycle analysis and RNAseq experiments, as shown in Figure 3. The decision to employ a concentration significantly above the IC50 for HeLa cells was made to ensure a pronounced and observable effect on cell cycle arrest and gene expression within the constrained timeline of these experiments. Using a higher concentration allowed us to capture the compound's immediate impacts on the cell cycle and to broadly assess its transcriptional effects, facilitating the identification of potential gene targets and pathways influenced by compound 34. This approach was deemed necessary to uncover the full spectrum of the compound's biological activity beyond its cytotoxic threshold. To improve clarity for our readers, we have now included a brief explanation in the manuscript detailing our rationale for selecting this higher concentration in these specific assays.
- Thank you for your insightful feedback on our DEGs analysis. We acknowledge the importance of comparing our results with non-treated HeLa and PC3 cells and with normal cells to fully understand compound 34's impact. Currently, limitations in data and resources constrained our ability to make these direct comparisons. In light of this, we have included a section in our discussion to carefully interpret our results within these constraints. We have endeavored to place our observations in the context of the existing literature, aiming to provide the reader with a clear understanding of what can be inferred from our findings given these limitations.
- Thank you for your feedback regarding our tubulin polymerization study and the specific concerns raised about the use of DMSO and the concentration of compound 34 and paclitaxel. We appreciate the opportunity to address these points to clarify our experimental approach. Regarding the concentration of paclitaxel used in our experiments, we opted for a 10 micromolar concentration in alignment with the guidelines provided by the tubulin polymerization assay kit (https://www.cytoskeleton.com/pdf-storage/datasheets/bk006p.pdf). This standard concentration was employed to clearly demonstrate the effect of paclitaxel on tubulin polymerization, serving as a benchmark for our study. Consequently, we applied the same concentration for compound 34 to maintain consistency and facilitate direct comparisons within the experimental framework. As for the concern about DMSO, we acknowledge that DMSO can indeed enhance tubulin polymerization in vitro, but at high concentrations. Our experimental setup utilized DMSO as a solvent for both compound 34 and paclitaxel, with its final concentration in the assay being within the range commonly used in such studies. We recognize, however, the importance of rigorously controlling for DMSO's effects to unequivocally determine its influence on the assay outcomes. While our current data did not explicitly include a control for the DMSO's polymerization effect at the concentrations used, we agree that incorporating such a control would strengthen the study's conclusions. This critical insight will be duly considered and implemented in future experiments to ensure a clearer interpretation of the results.
Reviewer 2 Report
Comments and Suggestions for Authors
The manuscript entitled “Identification of 3-aryl-1-benzotriazole-1-yl-acrylonitrile as a Microtubule-Targeted Agent (MTA) in solid tumors” describes the anticancer properties of 3-aryl-1-benzotriazole-1-yl-acrylonitrile (or compound 34), acting as promising microtubule-targeting agent in solid tumors. Several data and analysis are presented, confirming the ability to induce cell cycle arrest in the G2/M phase, disrupt tubulin polymerization, inhibit cell division, and activate the p53 signaling pathway.
The article is well organized and the results are interesting. Nevertheless, there are some issues in style of presentation and contents that should be addressed before publication.
- The abbreviation list is missing.
- In the introduction section (lines 60-61), any reference describing the increasing relevance of small molecules interfering with microtubules in cancer treatment is cited. Therefore, the following references should be added: https://doi.org/10.1016/j.ejcb.2020.151075; https://doi.org/10.1016/j.ejmech.2023.115372; https://doi.org/10.1002/mog2.46; doi: 10.37349/etat.2022.00112.
- In the Results section, some tests were superficially described. For example, for the colony formation assay, the choice of starting concentrations was not clear: if the idea was to test concentrations slightly above and below the IC50 obtained for the two different cell lines, authors should explain why the lowest compound concentration for HeLa cells (IC50 = 20 nM) was of 20 nM, while for PC3 cells (IC50 = 80 nM) it was of 50 nM.
- The implications of GO mapping and KEGG pathway analyses of DEGs should be more deeply discussed.
- The quality of Figures 3, 5, 6, 7 and 8 should be improved.
A revision of English is required, since few sentences should be rephrased. Moreover, some typos and misspelled words should be checked throughout the manuscript. Some examples are listed below:
- Page 1, lines 37-38: “Several cellular processes depend on microtubules, which by their nature are dynamic polymers.”
- Page 16, line 344: “Among these drugs are anti-mitotic drugs.”
- Page 16, lines 346-348: “In fact, α and β tubulin dimers fuse in a process of nucleation and polymerization with other dimers to form microtubular network and formation of the mitotic spindle, which is essential for the completion of the cell cycle.”
- Page 2, line 67: please remove the capital letter from the preposition "in" of the sentence “Therefore, In the current study its role to interfere with microtubulin function was explored…”.
- Page 4, line 165: please pay attention to the alignment of Table 1.
- Page 5, lines 167-175: the caption size of Table 1 should be reduced
- Page 15, line 317: please correct the paragraph title (it looks like "34" is the paragraph number).
For these reasons, in my opinion, the manuscript is suitable for publication in International Journal of Molecular Sciences after minor revisions.
Comments on the Quality of English LanguageModerate editing of English language required
Author Response
Reviewer #2:
- Thank you for noting the absence of an abbreviation list. Acknowledging its importance for clarity, we've added an abbreviation list to the manuscript's end, just before the references section. This should facilitate a better reading experience.
- Thank you for your suggestion to enrich our introduction with specific references regarding the role of small molecules in targeting microtubules for cancer treatment. We have now incorporated the recommended citations into the introduction, improving the context and supporting our discussion with relevant literature.
- We appreciate your observation on the description of our experimental approach in the Results section, specifically regarding the choice of starting concentrations for the colony formation assay. To clarify, the selection was based with the aim of studying the effects of the compound around IC50 values for HeLa and PC3 cells. The slight deviation in starting concentrations was driven by our intent to encompass a range that includes and extends slightly above and below the determined IC50 values, providing a comprehensive view of compound 34's efficacy. We have now added this explanation to the relevant section for greater clarity.
- Thank you for emphasizing the importance of a more detailed discussion on the GO mapping and KEGG pathway analyses of DEGs. Recognizing the value of these analyses in elucidating the biological significance and potential mechanisms underlying the effects of compound 34, we have expanded our discussion to delve deeper into the implications of our findings. This includes exploring how specific GO terms and KEGG pathways associated with the DEGs may contribute to understanding compound 34’s impact on cellular processes and pathways relevant to cancer treatment. We believe this enhanced discussion will provide readers with a more comprehensive view of the potential mechanisms by which compound 34 exerts its effects.
- Thank you for your detailed feedback on the language and presentation aspects of our manuscript. We have addressed each of your points as follows:
Page 1: We have revised the sentence to improve clarity: "Several cellular processes are reliant on the dynamic nature of microtubules, which are polymers capable of rapid assembly and disassembly."
Page 16: This sentence has been rephrased for better readability: "Among these, anti-mitotic drugs play a significant role."
Page 16: We have clarified the description of the process: "In this process, α and β tubulin dimers undergo nucleation and polymerization, joining with other dimers to form the microtubule network and the mitotic spindle, essential for cell cycle completion."
Page 2: The capitalization error has been corrected to: "Therefore, in the current study, its role in interfering with microtubulin function was explored..."
Page 4: We have adjusted the alignment of Table 1 to ensure it is correctly positioned within the manuscript layout.
Page 5: The caption size of Table 1 has been reduced for consistency with the rest of the document's formatting.
Page 15: The error in the paragraph title has been corrected to accurately reflect the content, ensuring that "34" is understood as part of the text and not as a paragraph number.
We appreciate your attention to these details, which has helped us improve the precision and quality of our manuscript.
Reviewer 3 Report
Comments and Suggestions for Authors
Please find attached document.

Editing of English language is required
Author Response
Reviewer #3:
- Thank you for your feedback regarding the colony formation assay. We understand the importance of methodological clarity and the value of providing comprehensive data. Regarding the inclusion of images at time zero (day 0) of the treatment, we note that it is not standard practice to include such images for colony formation assays in the publication phase. This approach aligns with methodologies employed in previously published scientific works, where the focus is on the end-point analysis to demonstrate the effect of the treatment on colony formation [1–3]. Therefore, we did not capture images at this specific time point, as our experimental design was aimed at assessing the end-point impact of compound 34 on colony growth.
Concerning the original images of the colony assay for Figure 2, we acknowledge the reviewer's point on the potential value of these images. While not typically required for publication, in the spirit of transparency and to facilitate further understanding of our results, we have decided to include these original images as supplementary material.
As for the suggestion to conduct additional experiments such as senescence assays, Western blot analysis for cell death markers, or flow cytometry (FACS with Annexin V and PI staining) to validate the conclusions drawn from the colony assay, we appreciate the importance of these experiments in providing a more detailed understanding of the mechanisms underlying the observed effects of compound 34. We are currently planning to undertake these analyses in a subsequent study that will delve deeper into the pathways involved and further elucidate the mechanisms by which compound 34 influences cancer cell biology. This forthcoming work aims to build upon the findings presented here, contributing to a more comprehensive understanding of compound 34's potential as a therapeutic agent.
- In response to your recommendation, we have revised Figure 2 to include data plotted in triplicate, with error bars representing standard deviations. Statistical analyses have been performed to ensure the reliability and significance of our findings, enhancing the figure's informative value.
- Acknowledging the reviewer's suggestion to include a heatmap for the DEG set comparison in Figure 4, we must clarify that, due to the specific focus of our initial analysis and constraints in data visualization resources at the time, a heatmap was not included in our original submission. Our primary aim was to highlight significant DEGs through traditional graphical representations, which we believed would succinctly convey our findings.
However, we recognize the value a heatmap brings in visually representing complex datasets, allowing for an intuitive grasp of gene expression changes over time. While we have not incorporated a heatmap in the current version of our manuscript, we are exploring advanced data visualization tools for future revisions and publications. This will enable us to include heatmaps and other sophisticated graphical representations to enhance the clarity and depth of our data presentation.
- In light of your feedback requesting a more detailed analysis of differentially expressed genes (DEGs), including the examination of key cell cycle regulators, DNA damage/repair markers, and apoptosis markers, we wish to provide additional context regarding the scope and focus of our current study. Our initial manuscript aimed to highlight the broad impact of compound 34 on cancer cell growth and cell cycle progression, based on our available data and experimental design.
While we understand the value of conducting comprehensive analyses, such as Western blotting to validate changes in specific markers and the activation of critical pathways like the p53 signaling pathway, our current resources and experimental setup were focused primarily on the initial identification and characterization of compound 34’s effects. As such, detailed validations of DEG changes through Western blot analysis or flow cytometry were beyond the scope of this initial investigation.
Consistent with our approach to the colony formation assay, where we emphasized endpoint analysis and included original images as supplementary material for transparency, we plan to explore the detailed mechanisms of action of compound 34 in future studies. This includes conducting the suggested additional experiments to validate the computational analysis and more deeply investigate the pathways involved in compound 34's action on cancer cells. Our future work will aim to provide the comprehensive data necessary to thoroughly understand the multifaceted mechanisms by which compound 34 exerts its effects, building on the foundation laid in the current study.
- Regarding the request for expression data on specific proteins such as TUBB4B, TUBB2A, TUBA1C, MCM5, UBC, CDK1, CCNE2, and MYC following treatment with compound 34, we must clarify that such detailed expression data are not currently available within our dataset. Our investigation into compound 34 primarily utilized broader assays focused on cell viability, cell cycle arrest, and apoptosis induction, rather than on the quantification of changes in individual protein expressions. We are actively seeking additional funding and collaborations to extend our investigations to include these critical protein expression analyses.
- The statement on lines 49-50 has been carefully reviewed and revised for accuracy. The revised statement clarifies the relationship between apoptosis, cell cycle arrest at the mitotic checkpoint, and the consequences of improper chromosome attachment or separation, ensuring the text reflects current scientific understanding.
Thank you for highlighting the need for a careful revision of the English language used in our manuscript. Following your feedback, we have meticulously reviewed and edited the manuscript to ensure that all sentences are phrased in correct and clear English. Our team has paid particular attention to correcting typos, misspellings, and any awkward phrasings throughout the document. This thorough internal review process has allowed us to enhance the manuscript's overall clarity and readability, ensuring that our scientific findings are communicated effectively.
- Franken, N.A.P.; Rodermond, H.M.; Stap, J.; Haveman, J.; van Bree, C. Clonogenic Assay of Cells in Vitro. Nature Protocols 2006 1:5 2006, 1, 2315–2319, doi:10.1038/nprot.2006.339.
- Bagella, L.; Sun, A.; Tonini, T.; Abbadessa, G.; Cottone, G.; Paggi, M.G.; De Luca, A.; Claudio, P.P.; Giordano, A. A Small Molecule Based on the PRb2/P130 Spacer Domain Leads to Inhibition of Cdk2 Activity, Cell Cycle Arrest and Tumor Growth Reduction in Vivo. Oncogene 2007 26:13 2006, 26, 1829–1839, doi:10.1038/sj.onc.1209987.
- Chen, Y.; Peng, C.; Chen, J.; Chen, D.; Yang, B.; He, B.; Hu, W.; Zhang, Y.; Liu, H.; Dai, L.; et al. WTAP Facilitates Progression of Hepatocellular Carcinoma via M6A-HuR-Dependent Epigenetic Silencing of ETS1. Mol Cancer 2019, 18, doi:10.1186/S12943-019-1053-8.
Round 2
Reviewer 3 Report
Comments and Suggestions for Authors
Thank you to the authors for their response and clarification regarding the comments. The writing in the manuscript has significantly improved. I have no further comments to add.
Author Response
We thank the reviewer again for the careful comments that helped improve the manuscript.